# Zinc Glycinate Alleviates Necrotic Enteritis Infection in Broiler Chickens

**DOI:** 10.3390/ani15162373

**Published:** 2025-08-13

**Authors:** Theros T. Ng, Brandi A. Sparling, Ramesh K. Selvaraj

**Affiliations:** 1College of Veterinary Medicine, Western University of Health Sciences, Pomona, CA 91766, USA; bsparling@westernu.edu; 2Department of Poultry Science, University of Georgia, Athens, GA 30606, USA; selvaraj@uga.edu

**Keywords:** chicken, zinc, necrotic enteritis

## Abstract

Necrotic enteritis is an intestinal infection in chickens, costing a significant economic loss globally. This disease is caused by a combination of coccidiosis and *Clostridium perfringens*. Therefore, it is essential to develop ways to mitigate the disease to ensure global food security. For this reason, we investigated dietary concentrations of zinc glycinate in chicken diets to alleviate necrotic enteritis infection using in vivo and in vitro approaches. We found that increasing zinc concentration to 120 mg/kg from the current recommendation of 100 mg/kg in the form of zinc glycinate successfully alleviated necrotic enteritis infection in chickens. This study provides the rationale to update the current recommendation for zinc in poultry diets.

## 1. Introduction

Necrotic enteritis is a costly intestinal infection in chickens that is caused by coccidiosis infection [1] and *Clostridium perfringens* [2]. Current efforts are aimed at improving the micronutrient status to improve immunity against the disease [3]. Zinc is an essential micronutrient for growth [4] and immunity [5] in chickens. Zinc (133 mg/kg) from organic sources improved body weight gain (BWG), feed conversion ratio (FCR), and modulated IL-8 and IL-10 cytokine expressions during infections better than inorganic sources (116 mg/kg) in one study [6]. However, the effective concentration of zinc glycinate in broiler chicken diets against necrotic enteritis is unclear. Despite the demonstrated efficacy of zinc in reducing damage by the pathogen, the expression of many immune genes (IL-1β, −6, 8, −10, IFN-γ, iNOS, and NF-kB) is unaffected by organic zinc (104 mg/kg) compared to inorganic zinc (97.4 mg/kg) in a subsequent study [7]. Importantly, previous studies have ignored zinc’s potential to exacerbate *Clostridial* proliferation in vitro [8]. The *Clostridium perfringens* Alpha (CPA) toxin of *C. perfringens* is a zinc metalloprotease [9], which can compete for the host’s micronutrient supply. Also, zinc decreases resistance to *C. difficile* infection in infants [10], an infection that can result in gastrointestinal symptoms. Therefore, it remains unclear whether zinc reduces necrotic enteritis by modulating the immune responses of chickens or the proliferation of *C. perfringens* in the intestine.

The objective of this study was to determine the effective concentration of zinc glycinate in broiler chicken diets to alleviate necrotic enteritis and to better determine the molecular underpinnings surrounding zinc in infections. We supplemented zinc glycinate between 40 mg/kg and 160 mg/kg in the diets and experimentally induced necrotic enteritis with *Eimeria* and *C. perfringens* [11]. We hypothesize that zinc glycinate above the current recommendation at 100 mg/kg [12] reduces gross lesions from necrotic enteritis and improves growth performance after the infection by modulating cellular and humoral immunity. Primary cells, especially intestinal cells, are difficult to culture because of sloughing, high cell turnover, and the high number of bacteria. We developed this novel in vitro method to test the effects of zinc concentration and source (sulfate and glycinate) within primary chicken intestine cells or a chicken macrophage cell line (HD-11) co-cultured with *C. perfringens*.

## 2. Materials and Methods

### 2.1. Animals and Husbandry

The animal protocol was approved by the Institutional Animal Care and Use Committee (IACUC) at the University of Georgia under protocol A2018 04-010-Y2-A2. Two experiments were conducted at Southern Poultry Research Inc. (Athens, GA, USA) to investigate the effects of zinc glycinate concentrations on immunity. A priori power analysis utilizing the effect size from a similar study of selenium glycinate on the immune status of chickens [3] was used to determine the sample size required to achieve 97% power (G*Power ver. 3.0.10) [11]. Birds were housed under standard animal husbandry practices in Petersime battery cages, and water was provided ad libitum. The institutional animal care and use committee (IACUC—Southern Poultry Research, Athens, GA, USA) approved our experimental protocols. Day-of-hatch male Cobb 500 broiler chicks (Cobb-Vantress, Cleveland, GA, USA) were randomly assigned to the zinc glycinate treatments. Each treatment was replicated in six Petersime battery cages with eight birds per cage (n = 6). Each cage was an experimental unit. The floor space per animal was 0.06 m^2^/bird. The feeder/water space per bird was eight birds per feeder/water trough. A thermostatically controlled gas furnace/air conditioner was used to maintain a uniform temperature. Illumination was also provided. The experimental diets were supplemented with final zinc concentrations from zinc glycinate at 40, 80, and 120 mg/kg in experiment 1 and 100, 120, 140, and 160 mg/kg in experiment 2. A total of 192 and 240 birds were used in experiments 1 and 2, respectively. Each treatment group included six biological replicates, with eight birds per replicate (48 birds per treatment). The molecular weight of zinc in the zinc glycinate production was used to calculate the amount needed to achieve the final zinc concentrations. The experimental diets were prepared from a basal unmedicated starter diet consisting of a zinc-free mineral mix. The composition of the diet, except for zinc, was based on the Cobb 500 Performance and Nutrition Supplement Guide [12] (Table 1). The test diets were iso-caloric and iso-nitrogenous.

### 2.2. Experimental Necrotic Enteritis Induction

The necrotic enteritis infection model used in both experiments has been previously validated [6,7]. Necrotic enteritis was induced by oral gavage of 5000 oocysts/mL of *Eimeria maxima* (field isolates from Dr. Lorraine Fuller at the University of Georgia, Athens, GA, USA) on day 14 and 108 CFU/mL of *Clostridium perfringens* daily on days 19, 20, and 21 [13,14]. The *C. perfringens* was an isolate provided by Dr. Charles Hofacre at the Southern Poultry Research Group, Inc. (Athens, GA, USA), from a clinical case of necrotic enteritis and was positive for both alpha toxin and netB toxins. Separately, birds (n = 6) fed 40 mg/kg in experiment 1 and 100 mg/kg in experiment 2 were uninfected to ensure the efficacy of the infection model.

### 2.3. Sample Collection

On day 21, a subset of birds (n = 6) was euthanized. The jejunal tissues were collected for histology in experiment 1. Cecal tonsils, spleen, and jejunal tissues were snap-frozen in liquid nitrogen for gene expression analyses. Cecal content was collected for DNA extraction and microbial 16S, *C. perfringens*, cpa, and *netB* analysis via RT-PCR. Gross lesions (0: lowest to 3: highest) were evaluated on day 21 by a trained pathologist who was blinded to treatments [15]. On day 28, the remaining birds were analyzed for growth performance recovery after the infection. Birds and feed were weighed on days 0, 14, 21, and 28. Body weight and feed intake were measured on a per-cage (replicate) basis. Feed intake was calculated by subtracting the weight of feed refusals from the total feed provided. Feed conversion ratio (FCR) was calculated for each replicate as the total feed intake divided by the total body weight gain, and averaged per bird within each cage.

### 2.4. Histology

In experiment 1, the jejunum tissue was fixed in 10% neutral buffered formalin and processed at the Poultry Diagnostic and Research Center (PDRC, Athens, GA, USA) laboratory. The cross-sections were viewed and photographed using an Olympus IX71 microscope and analyzed using a DP Controller (ver. 2.1.1.183 software) to determine the villi length and crypt depth. Five villi per section and four sections per sample were analyzed for a total of 120 samples. For each jejunum sample, villi and crypts of four sections of the slides were measured and averaged as technical replicates. The ratios of villi length to crypt depth were calculated.

### 2.5. Real-Time PCR Gene Expression

Total RNA from the cecal tonsil, spleen, and jejunum tissues was isolated using the TRIzol/chloroform method. Briefly, tissues (approximately 25 mg) were homogenized using the TissueLyser LT (Qiagen, Germantown, MD, USA) and 5 mm stainless steel beads in 1 mL TRIzol for 6 min. Chloroform (200 µL) was added to the homogenate and centrifuged to fractionate the total RNA into the aqueous phase. The total RNA was precipitated in isopropanol, washed in 70% ethanol, and then resuspended in molecular-grade water. Optical absorbance at 260 nm was used to determine RNA concentrations. RNA samples were reverse-transcribed. Two micrograms of RNA were converted to cDNA in a 20 μL reaction volume containing 1× reaction buffer (50 mM Tris–HCl [pH 8.3], 75 mM KCl, 3 mM MgCl2, and 10 mM DTT), 10 mM DTT, 0.5 mM dNTPs, 0.5 μM of oligo(dT)15 primer, 8 units of RNAsin, and 100 units of M-MLV reverse transcriptase (all from Promega) at 40 °C for 1 h and then 95 °C for 10 min. Samples were stored at −20 °C before use.

In experiments 1 and 2, relative gene expressions of pro-inflammatory cytokines (IL-1β, IFN-γ, and LITAF), anti-inflammatory cytokines (IL-10 and TGF-β) of mRNA from cecal tonsil and spleen tissues were analyzed using the CFX-96 and CFX Maestro Software (Bio-Rad, Hercules, CA, USA). Relative gene expressions of jejunum tight junction proteins (claudin-2 and occludin) and zinc importer (ZIP-9) were analyzed. In experiment 2, Zn/Cu-SOD-1 gene expressions were measured in the cecal tonsils and spleen. The expressions were normalized to β-actin as the housekeeping gene and relative to expressions of the zinc glycinate at 40 mg/kg infected treatment in experiment 1 and zinc glycinate at 100 mg/kg infected treatment in experiment 2. Primers and cycle conditions are described in Table 2. Gene expressions are reported as fold-change ± SEM. For all reactions, we used the PerfeCTa SYBR Green FastMix (cat# 95072, Quantabio, Beverly, MA, USA) according to the manufacturer’s directions. Fold-change from the reference was quantified as 2^−∆∆Cq^ [16], where Cq is the threshold cycle, defined as the cycle number when the product fluorescence reaches exponentiality above background noise.

### 2.6. Cecal Microbial Analysis Using RT-PCR

In experiment 2, cecal contents were homogenized in sterile bags. DNA was extracted using a commercial kit (QIAamp DNA Stool Mini Kit, Qiagen Hilder, Germany) according to the manufacturer’s directions. The *Clostridium perfringens*, *netB*, and *cpa* genes were quantified using the primers and cycle conditions described in Table 3. Expressions were normalized to 16S rRNA and relative to the zinc glycinate at 100 mg/kg infected treatment. The relative fold change was calculated as described above.

### 2.7. Culturing of Primary Intestinal Epithelial Cells with Zinc

A modifed intestinal ex vivo method was performed to evaluate the effect of zinc [24]. The jejunum was collected from three broiler chickens (n = 3) at 28 days of age and suspended in complete media (DMEM media containing 10% FBS, 1% penicillin-streptomycin, and 2 mM glutamine). The tissues were washed with PBS until the PBS ran clear and then minced into 5 cm sections for enzymatic digestion. Next, the tissue was allowed to incubate in collagenase (0.3 mg/mL in DMEM) for 15 min and washed with DMEM through a 70 µm cell strainer. This was repeated three to four times [25]. The fourth and fifth digestions containing the intestinal crypt were filtered through a 70 μm cell strainer. The remaining collagenase was washed away by centrifugation in PBS for 8 min at 400× *g*.

The LDH assay was performed to examine for any cytotoxicity of the cells. The primary chicken intestinal cells were seeded in 24-well plates at approximately 80% confluency in complete media as control or complete media containing zinc at 10 or 100 μM supplementation from zinc sulfate or zinc glycinate for 12 or 24 h (41 °C and 5% CO_2_), after which the supernatant was collected for LDH assay. Briefly, the phenazine methosulfate (PMS), ioinitotetrazolium chloride (INT), and diphosphopyridine nucleotide (NAD) (1:1:23 *v*/*v*/*v*) were mixed before the assay. Then, the supernatant was diluted in the LDH reagent (1:1:1 *v*/*v*/*v* of 200 mM TRIS at pH 8, 50 mM lithium lactate, and PMS, INT, and NAD mixture) in a 1-to-3 *v*:*v* ratio. Maximum release was relative to cells lysed with 1% Triton X-100 (*v*/*v*). Absorbance was read in the microplate reader at OD490.

The MTT assay was performed to assess the relative cell viability of the cells. Primary chicken intestinal cells were seeded in 96-well plates at approximately 80% confluency in 100 μL phenol-red free RPMI media containing 10% FBS and 1% penicillin-streptomycin as control or 100 microliters control media containing zinc at 10 or 100 μM from zinc sulfate or zinc glycinate for 24 h (41 °C and 5% CO_2_). The MTT reagent (20 μL) was added directly to the wells and allowed to incubate for 4 h in the dark to accumulate formazan product, and then, 100 μL DMSO was added to the wells to dissolve the product overnight in an incubator set to 41 °C. The absorbance was read at OD575 in the microplate reader.

### 2.8. Culturing of Chicken Macrophage Cell Line (HD-11) with Zinc

The HD-11 experiment was repeated, resulting in two biological replications (n = 2). HD-11 cells were cultured at 41 °C and 5% CO_2_ in complete media (DMEM media containing 10% FBS and 1% penicillin-streptomycin). *C. perfringens* (CP6) was thawed, washed with PBS to remove DMSO, and grown in TSB-T medium overnight. The bacteria were in the late log to the stationary phase before being washed thrice in PBS through centrifugation at 3000 rpm for 10 min. The bacteria were resuspended in complete media, thew OD600 absorbance was read, and the bacteria were added to the appropriate wells at the multiplicity of infection (MOI) of 10 bacteria to one cell.

For the NO assay, a total of 50,000 cells were seeded in 96-well plates in complete media or complete media containing zinc at 10 or 100 μM from zinc sulfate or zinc glycinate 10 or 100 μM for 24 h. Subsequently, an MOI of 10 bacteria to one cell was added and incubated for 2 h (41 °C and 5% CO_2_) before extracellular bacteria were rinsed off with PBS twice. Finally, cells were incubated for 24 h in phenol-red-free RPMI media containing 10% FBS and 1% penicillin-streptomycin to allow the accumulation of nitric oxide species. After, an equal volume of Griess reagent was added to supernatants, and the absorbance was read with a microplate reader at OD540. The relative concentrations of nitrate were compared to that of a sodium nitrate standard curve.

For the LDH assay, a total of 250,000 cells were seeded into 24-well plates in complete media or complete media containing zinc at 10 or 100 μM from zinc sulfate or zinc glycinate for 24 h. Subsequently, an MOI of 10 bacteria to one cell was added and incubated for 2 h (41 °C and 5% CO_2_) before extracellular bacteria were rinsed off with PBS twice. Cells were incubated for 24 h in complete media before the supernatant was collected for LDH determination.

For the visualization of phagocytosed bacteria, after removal of media, cells were fixed with methanol for 1 min and then subsequently stained with Giemsa for 1 min. Then, immediately following, the excess stain was removed by gentle rinsing in PBS.

### 2.9. Statistical Analysis

Growth performance (BWG, FI, and FCR), and gene expression data were analyzed using a one-way ANOVA in SAS (v.9.0, SAS Institute, Inc., Cary, NC, USA) to determine the effects of zinc glycinate concentrations on the dependent variables. When the main effect was significant (*p* < 0.05), the differences between means were analyzed using Tukey’s HSD post hoc analysis. Intestinal lesion scores were analyzed by a Kruskal–Wallace non-parametric test and post hoc Tukey’s HSD. Mortality was analyzed by the chi-square test and Mann–Whitney U-test post hoc.

A two-way ANOVA (JMP Pro 15) was used to examine the main effects and interaction of zinc source and concentration on the dependent variables in the in vitro assays. When the main effects were significant (*p* < 0.05), differences between means were determined using Tukey’s HSD.

## 3. Results

The following results report only the comparisons between the infected groups, as the uninfected control only served to ensure the necrotic enteritis infection model.

### 3.1. Effects on Growth Performance

On days 21 (Table 4) and 28 (Table 5) in experiment 1, body weight gain, feed intake, and feed conversion ratio were not significantly different between the infected birds fed different zinc glycinate concentrations.

On day 21 (Table 4) of experiment 2, body weight gain, feed intake, and feed conversion ratio were not significantly different between the infected birds fed different zinc glycinate concentrations. On day 28 in experiment 2, feed intake was not significantly different between treatments.

During the recovery phase on day 28 (Table 5) in experiment 2, infected birds supplemented with zinc glycinate at 120 mg/kg had a higher body weight gain by 303 g/bird compared to infected birds supplemented at 100 mg/kg (*p* < 0.05). At the same time, infected birds supplemented with zinc glycinate at 140 mg/kg had a lower feed conversion ratio by 0.81 compared to infected birds supplemented at 100 mg/kg (*p* < 0.05).

### 3.2. Effects on Gross Lesions and Mortality

On day 21 in experiment 1, infected birds supplemented with zinc glycinate at 120 mg/kg yielded a lower lesion score than those at 80 mg/kg (*p* < 0.01) (Table 4). On day 21 in experiment 2, the lesion score was lower in infected birds supplemented with zinc glycinate at 120, 140, and 160 mg/kg than in infected birds supplemented at 100 mg/kg (*p* < 0.01) (Table 4). On day 21 in experiment 1, mortality was lower in infected birds supplemented with zinc glycinate at 80 mg/kg than in infected birds supplemented at 40 mg/kg (*p* < 0.01) (Table 4). On day 21 in experiment 2, morality was lower in infected birds supplemented with zinc glycinate at 120 mg/kg than in infected birds supplemented at 100 mg/kg (*p* < 0.01) (Table 4).

### 3.3. Effects on Jejunum Histological Parameters

On day 21 in experiment 1, zinc glycinate concentration had no significant effects on villi length, crypt depth, and ratio of villi to crypt depth (Table 6).

### 3.4. Effects on Tissue Relative mRNA Expressions

The cytokine gene expressions of cytokines (IFN-γ, LITAF, and IL-10), tight junction proteins (claudin-2 and zona occluden), and zinc transporter protein (ZIP-9) for experiment 1 (Figure 1) and experiment 2 (Figure 2) are reported. Each bar represents the mean ± SEM of six replicates of birds (n = 6). The mRNA expressions are expressed as fold-change normalized to β-actin and relative to the uninfected group with zinc glycinate 40 mg/kg diet. Values with different letters represent significant differences in means. Only the expression levels that are significant among the infected groups are reported here.

On day 21 in experiment 2, the expression of zinc/copper superoxide dismutase-1 gene (Zn/Cu SOD-1) in the cecal tonsils was significantly lower in birds supplemented at 140 mg/kg compared to other supplemental concentrations (100, 120, and 160 mg/kg) (*p* < 0.05). On day 21 in experiment 2, the expression of IFN-γ mRNA in the spleen was significantly higher in birds supplemented at 160 mg/kg compared to that at 100 mg/kg (*p* < 0.01).

### 3.5. Effect on C. perfringens Gene Expression

On day 21 in experiment 2, *C. perfringens*, *netB*, and *cpa* expressions in the ceca were not statistically different between treatments (Figure 3).

### 3.6. Zinc Co-Culture with Primary Intestinal Cells

In the chicken intestinal epithelial cells, the addition of zinc at 10 or 100 μM resulted in significantly higher MTT production (Figure 4a) (*p* < 0.05). The main effect, zinc concentration, significantly affected intestinal cells’ LDH production at 12 h (*p* < 0.001) and 24 h (*p* < 0.001). LDH production from the intestinal cells at 12 h after incubation was significantly lower in the treatments supplemented with zinc at 100 μM (*p* < 0.05) (Figure 4b). LDH production from the intestinal cells 24 h after incubation was significantly lower in the treatments supplemented with zinc at 10 μM. Images of the jejunal crypt cells under different conditions showed less cell confluency under 100 μM of zinc, regardless of the source (Figure 4c).

### 3.7. Zinc Co-Culture with HD-11 Cells

In HD-11 cells, LDH production at 24 h decreased as zinc concentrations increased (*p* < 0.01) (Figure 5a). Infection LDH production from the intestinal cells at 12 h after incubation was significantly lower in the treatments supplemented with zinc at 100 μM (*p* < 0.05) (Figure 4b). LDH production from the intestinal cells 24 h after incubation was significantly lower in the treatments supplemented with zinc at 10 μM. Images of the jejunal crypt cells under different conditions showed less cell confluency under 100 μM of zinc, regardless of the source (Figure 4c).

### 3.8. Effect of Zinc on Clostridium perfringens Growth

*Clostridium perfringens* growth was significantly suppressed by zinc regardless of form (Table 7 and Figure 6). After 2 h of culturing the bacteria with zinc, bacterial growth was suppressed above 100 μM. Throughout the study, bacterial growth was not different between control (0 μM), zinc sulfate at 10 μM, and zinc glycinate at 0 μM. At the 2 h timepoint, zinc sulfate and glycinate inhibited bacterial growth at 1000 μM (*p* < 0.001). In addition, zinc glycinate lowered bacterial growth more significantly than zinc sulfate (*p* < 0.01). At the 4 h timepoint, zinc sulfate and glycinate lowered bacterial growth at 100 μM (*p* < 0.001). Bacterial growth was the lowest with zinc glycinate at 1000 μM (*p* < 0.001). At the 4 h timepoint, zinc sulfate and glycinate lowered bacterial growth at 100 μM (*p* < 0.001). Bacterial growth was the lowest with zinc glycinate at 1000 μM (*p* < 0.001).

## 4. Discussion

Zinc is an essential micronutrient that serves as a cofactor in various biological processes [4]. For example, the antioxidant enzyme copper/zinc superoxide dismutase uses zinc as a cofactor to reduce oxidative stress [26]. Zinc is also involved in the structural formation of zinc-finger binding domains in transcription factors such as RANTES [27], Th-POK [28], and Egr-2 [29], which are important for T-cell activation. Zinc glycinate, an organic form of zinc, has been used in chickens to alleviate damage from pathogens such as coccidia [30], necrotic enteritis [31], *Salmonella enteritidis* [32,33], and *Campylobacter jejuni* [34].

This study examined the effects of zinc in the form of zinc glycinate on necrotic enteritis in broiler chickens, a disease primarily caused by the overgrowth of *Clostridium perfringens* in the intestine [35,36]. Clostridial toxins, particularly CPA and netB, cause damage to the intestinal mucosa, leading to necrosis [37]. Previous research demonstrated that zinc from organic sources, such as zinc proteinate, reduces necrotic enteritis lesions by modulating IL-8 and IFN-γ cytokine expression [7].

The goal of this study was to determine the effective concentration of supplemental zinc glycinate above the current guideline of 100 mg/kg to reduce necrotic enteritis. This study omitted uninfected controls for higher zinc glycinate treatments because our only interest is the effects during a necrotic enteritis infection. Furthermore, cytokine expressions remain relatively unaffected in healthy birds below 100 mg/kg, which is the current guideline for poultry [38]. Uninfected controls at 40 mg/kg were only used to monitor if the infection model was compromised (i.e., uninfected controls showing signs of an infection). For this purpose, neither the uninfected controls showed any sign of necrotic enteritis infection in either experiment.

We conducted the first study investigating zinc during necrotic enteritis around the current guideline of 100 mg/kg. But we found that the effects were minimal, except that higher zinc concentrations reduced lesion score and mortality. Therefore, the concentration of zinc in the subsequent study was increased at levels exceeding the current guideline. On that note, zinc glycinate was suggested to have higher bioavailability than its inorganic counterparts [39,40,41,42]. Therefore, even at 100 mg/kg, zinc glycinate should theoretically provide more nutrients to the animals. However, our scope was not to compare zinc sulfate and zinc glycinate during the infection; therefore, we limited our study design to only include zinc glycinate in our treatments. In our subsequent study, zinc glycinate at 120 mg/kg reduced gross intestinal lesions and mortality in the birds, with a 13% reduction in mortality observed in Experiment 2. Following the infection on day 21, zinc glycinate at 120 mg/kg improved body weight gain, and zinc glycinate at 140 mg/kg improved the feed conversion ratio in the remaining birds that survived the infection on day 28. The current dietary inclusion concentration of zinc is between 100 mg/kg [12] to 120 mg/kg [43]. These results suggest that zinc glycinate, particularly at 120 mg/kg, helps alleviate necrotic enteritis and supports growth recovery. Future recommendations for poultry diets should consider this, especially if the goal is to optimize immunity against diseases. Considering that the cost of 20 milligrams of zinc per kilogram of poultry feed is minuscule, the benefit of the increase can translate to a dramatic gain in case of an infection. Despite the benefit we found by increasing zinc glycinate from 100 to 120 mg/kg, a larger-scale study would be needed to demonstrate the monetary benefit.

Another goal of this study was to elucidate the immune pathways that can explain the reduction in necrotic enteritis by zinc glycinate. A previous study observed slight differences in cytokine expression in birds supplemented with organic zinc [7]. Unfortunately, we found no significant changes in cytokine expression except for IFN-γ in the spleen during experiment 2. Many factors modulate cytokine expressions; therefore, selectively analyzing individual cytokines may not reflect the immune status of the birds. Therefore, a more comprehensive analysis of the immune landscape, such as transcriptomic profiling, may be necessary to better understand the broader inflammatory status of the birds after supplementation [44]. Despite the expected reduction in IFN-γ in response to reduced infection severity, this was not observed. This suggests that the previous observation of IFN-γ may or may not be related to the challenge. We noticed improved viability of the intestinal cells in the presence of zinc, even as low as 10 µM. Cytotoxicity of the intestinal cells was also lowered by zinc. Our in vitro assays seem to indicate that zinc glycinate enhanced cellular proliferation and robustness overall, though zinc may not be enhancing particular cytokine responses.

Zinc glycinate at 140 mg/kg reduced zinc/copper superoxide dismutase gene expression in the cecal tonsils post infection. Superoxide dismutase is crucial for reducing oxidative stress in chicken macrophages [45], and other micronutrients like manganese have been shown to modulate this enzyme [46]. This indicates that zinc glycinate may play a more significant role in innate immunity during necrotic enteritis. Possible connections include whether this relates to maintaining homeostatic balance, whether zinc influences dismutase enzyme sensitivity, and whether zinc plays a role in non-classical mechanisms.

Zinc is primarily absorbed in the jejunal brush border [47]. Despite reducing gross lesions, zinc glycinate at 120 mg/kg did not affect jejunal villi morphology or the expression of tight junction proteins such as claudin-2 and zonula occludens. The study also found no effect on jejunal ZIP-9 transporter expression, a transporter involved in zinc transport in the trans-Golgi network [48,49]. Therefore, there is no reason at this time to suggest that the rate of zinc uptake, which may be distinct from bioavailability, has otherwise changed.

This study focused on the early growth phase and disease impact, concluding at day 21 to capture the acute effects of necrotic enteritis and early performance outcomes. While broilers are commonly raised to day 42, early slaughter is not uncommon in commercial operations, particularly under disease pressure. Future studies are warranted to assess the long-term effects of zinc supplementation on performance and health through to market age. Despite the potential benefits of zinc glycinate to the chicken, excess trace minerals can cause a detrimental effect because bacteria also utilize trace minerals for their metabolism. Particularly, zinc is a cofactor for metalloproteases that contribute to the virulence of *Clostridium perfringens* [50]. To demonstrate that zinc does not promote the growth of *C. perfringens*, we cultured the bacteria with different concentrations of zinc sulfate and zinc glycinate. An increase in zinc, regardless of the zinc source, did not increase *C. perfringens* growth. Therefore, dietary zinc is unlikely to promote *C. perfringens* proliferation in the intestine directly. Next, we co-cultured primary chicken jejunal cells with zinc to determine the direct effects of zinc on cellular responses. We discovered that the zinc source was not an important factor. However, zinc improved viability and reduced basal cytotoxicity under in vitro conditions.

## 5. Conclusions

Increasing dietary zinc concentration from 100 to 120 mg/kg in the form of zinc glycinate significantly improved outcomes in chickens challenged with necrotic enteritis. This supplementation led to reduced intestinal lesion severity, improved growth performance, and decreased mortality. In vitro assays further demonstrated that zinc inhibited *Clostridium perfringens* proliferation and supported intestinal cell growth. Based on these findings, zinc glycinate at 120 mg/kg is recommended as a practical dietary intervention to mitigate necrotic enteritis in poultry, particularly under conditions where disease risk is elevated.

## Figures and Tables

**Figure 1 animals-15-02373-f001:**
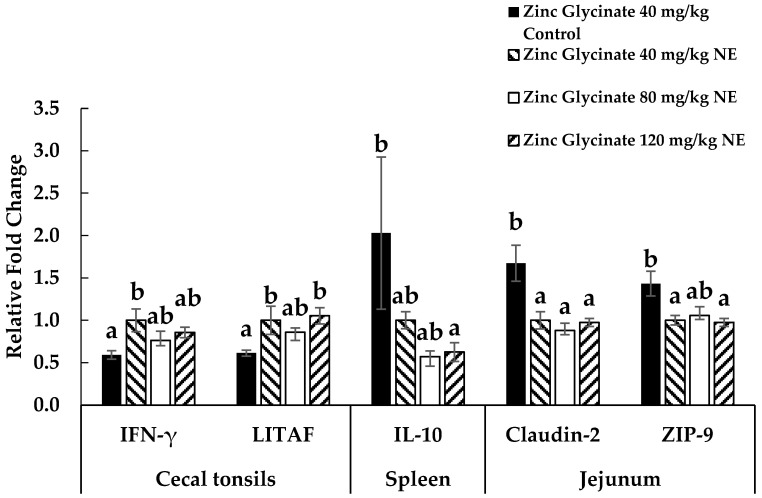
Effects of supplementation of dietary zinc glycinate at 40, 80, and 120 mg/kg on gene expression of cytokines in the cecal tonsils (IFN-γ and LITAF), spleen (IL-10), and jejunum (tight junction protein claudin-2 and zinc transporter ZIP-9) on day 21 in experiment 1. Each bar represents the mean ± SEM of six replicates. The mRNA expression is presented as fold-change, normalized to 16S rRNA as the housekeeping gene expression and to zinc glycinate 40 mg/kg NE. ^a,b^ Means carrying different superscripts are significantly different at *p* < 0.05.

**Figure 2 animals-15-02373-f002:**
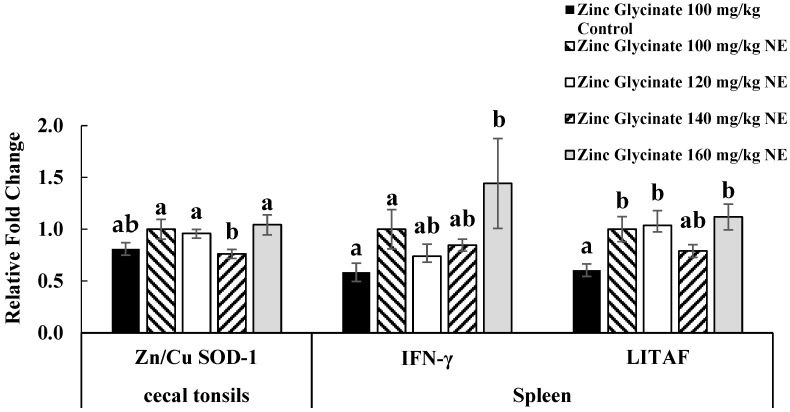
Effects of supplementation of dietary zinc glycinate at 100, 120, 140, and 160 mg/kg on gene expressions of antioxidant protein in the cecal tonsils (Zn/Cu SOD-1) and cytokines in the spleen (IFN-γ and LITAF) on day 21 in experiment 2. Each bar represents the mean ± SEM of six replicates. The mRNA expression is presented as fold-change, normalized to 16S rRNA as the housekeeping gene expression and to zinc glycinate 100 mg/kg NE. ^a,b^ Means carrying different superscripts are significantly different at *p* < 0.05.

**Figure 3 animals-15-02373-f003:**
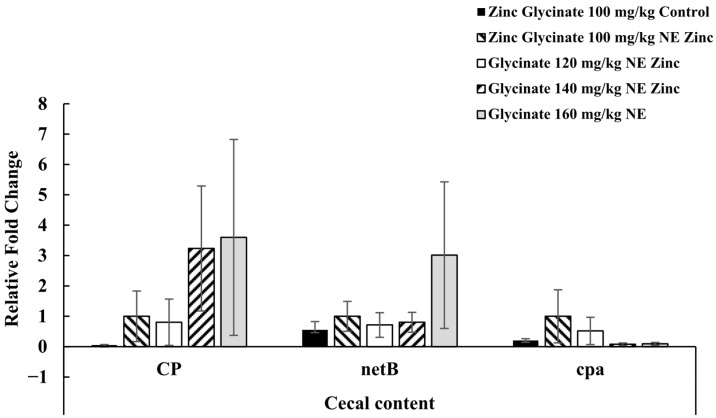
Effects of supplementation of dietary zinc glycinate at 100, 120, 140, and 160 mg/kg on gene expressions of cecal bacteria genes (*Clostridium perfringens* (CP), *netB*, and *cpa*) on day 21 in experiment 2. Each bar represents the mean ± SEM of six replicates. The mRNA expression is presented as fold-change, normalized to 16S rRNA as the housekeeping gene and calibrated to zinc glycinate 100 mg/kg NE. There were no significant differences between treatments (*p* > 0.05).

**Figure 4 animals-15-02373-f004:**
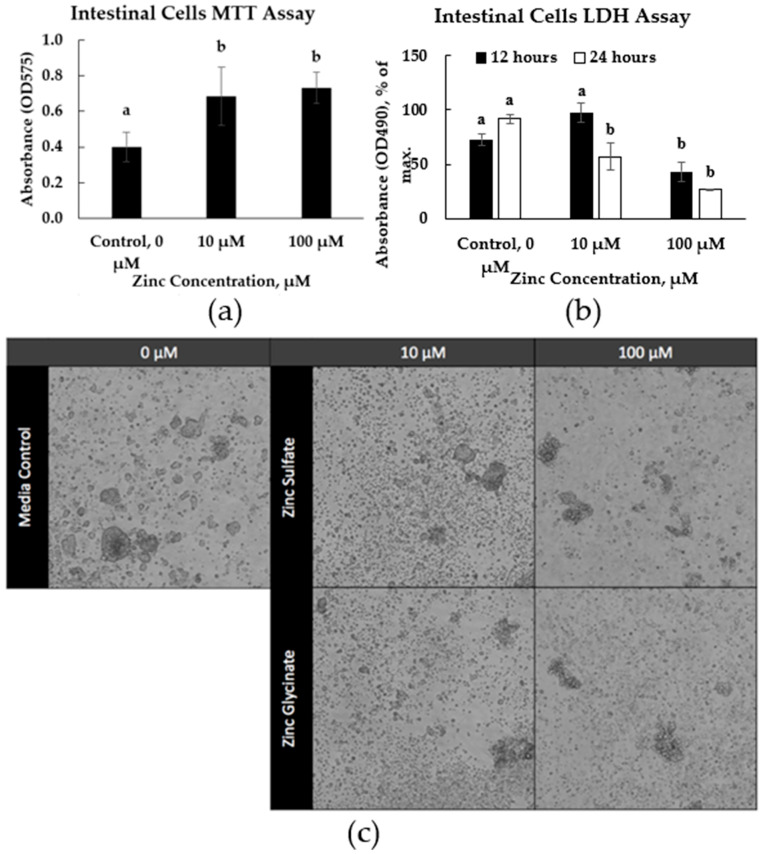
Viability and cytotoxicity of chicken jejunum intestinal cells exposed to zinc. (**a**) The main effect, zinc concentration on cell viability, was measured by the MTT assay. ^a,b^ Means within the same column carrying different superscripts are significantly different at *p* < 0.05. (**b**) The main effect, zinc concentration on cell cytotoxicity, was measured by the LDH assay. (**c**) Images of jejunal crypt cells cultured without or with zinc (0, 10, or 100 μM) from zinc sulfate or glycinate. ^a,b^ Means carrying different superscripts are significantly different at *p* < 0.05.

**Figure 5 animals-15-02373-f005:**
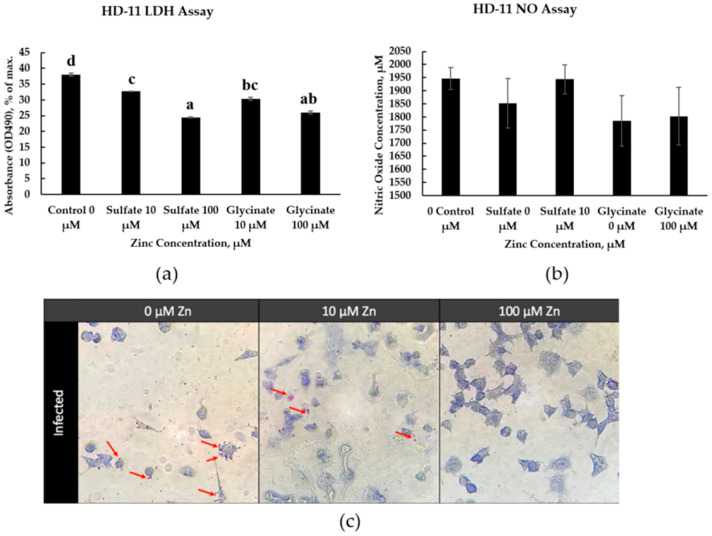
HD-11 cell cytotoxicity and NO production. (**a**) Effects of zinc concentration on cell cytotoxicity of HD-11 cells, measured by the MTT assay. (**b**) Effects of zinc concentration on nitric oxide production of HD-11 cells, measured by the NO assay using Griess reagent. (**c**) Images of HD-11 cells infected with *C. perfringens* during co-culturing with zinc at 0, 10, or 100 μM. Red arrows depict the presence of bacteria captured by the macrophages. ^a–d^ Means within the same column carrying different superscripts are significantly different at *p* < 0.05.

**Figure 6 animals-15-02373-f006:**
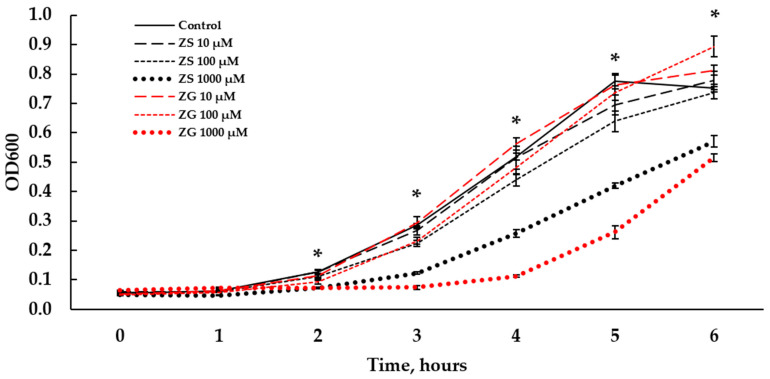
*Clostridium perfringens* growth curves as measured by OD at 590 nm during culture without or with zinc (0, 10, 100, or 1000 μM) from zinc sulfate or zinc glycinate over six hours. Asterisks describe significant differences between treatments within each timepoint at *p* < 0.01.

**Table 1 animals-15-02373-t001:** Dietary formulations, expected nutrient compositions, and ME for treatments based on formulation.

		Basal
Item	Ingredient or Nutrient (As-Fed Basis)	(1 to 28 Days)
Dietary inclusion (%)	Yellow corn grain	62.82
	Soybean meal (48%)	33.27
	Soybean oil	1.17
	Defluorinated phosphate	1.13
	Calcium carbonate	0.75
	^†^ Vitamin premix	0.25
	DL-Methionine	0.24
	L-Lysine	0.16
	Salt	0.12
	^‡^ Trace Mineral (Zinc-free)	0.08
	Quantum Blue Phytase (5000 FTU^§^/g)	0.01
Expected nutrient composition (%) and ME (kcal/kg) based on calculations	Dry matter	87.73
	Crude protein	21
	Crude fat	3.69
	Crude fiber	2.19
	Calcium	0.9
	Total phosphorus	0.58
	Available phosphorus	0.45
	Metabolizable energy	3008
	Methionine	0.58
	Lysine	1.28
	Tryptophan	0.28
	Threonine	0.85
	Sodium	0.16
	Potassium	0.82
	Chloride	0.15
	Digestible methionine	0.56
	Digestible cysteine	0.32
	Digestible lysine	1.18
	Digestible tryptophan	0.27
	Digestible threonine	0.77
	Digestible isoleucine	0.96
	Digestible histidine	0.53
	Digestible valine	1.05
	Digestible leucine	1.75
	Digestible arginine	1.33
	Digestible phenylalanine	1.07
	Digestible TSAA ^†^	0.88

^†^ Vitamins and minerals were provided in the form and amount described in the Cobb 500 Broiler Performance and Nutrition Supplement guide. Vitamin A: 10,000 IU; vitamin D3: 5000 IU; vitamin E: 8000 IU; vitamin K3: 3 mg/kg; vitamin B1 (thiamine): 3 mg/kg; vitamin B2 (riboflavin): 9 mg/kg; vitamin B6 (pyridoxine): 4 mg/kg; vitamin B12: 0.02 mg/kg; biotin: 0.15 mg/kg; choline: 500 mg/kg; folic acid: 2 mg/kg; nicotinic acid: 60 mg/kg; pantothenic acid: 15 mg/kg. ^‡^ Manganese (from manganese sulfate): 100 mg/kg; iron (from ferrous sulfate): 40 mg/kg; copper (from copper sulfate): 15 mg/kg; iodine (from calcium iodide): 1 mg/kg; selenium (from sodium selenite): 0.35 mg/kg. ^§^ FTU: phytase unit.

**Table 2 animals-15-02373-t002:** Primer information for semi-quantitative real-time PCR of gene expressions in the cecal tonsil, spleen, and jejunum tissues.

Gene		Primer Sequence (5′–3′)	Accession Number/Reference	Annealing	Extension
Housekeeping	*β* *−actin*	F: ACCGGACTGTTACCAACACC	[17]	57.5 °C, 30 s	10 s
		R: GACTGCTGCTGACACCTTCA			
Pro-inflammatory Cytokine	*IL-1β*	F: TGGGCATCAAGGGCTACA	Y07922/[18]	57.5 °C, 45 s	20 s
		R: TCGGGTTGGTTGGTGATG			
	* IFN-γ *	F: GTGAAGAAGGTGAAAGATATCATGGA	[17]		
		R: GCTTTGCGCTGGATTCTCA			
	* LITAF *	F: ATCCTCACCCCTACCCTGTC	[19]		
		R: GGCGGTCATAGAACAGCACT			
Anti-inflammatory Cytokine	*TGF-β*	F: CGGGACGGATGAGAAGAAC	M31160/[18]		
		R: CGGCCCACGTAGTAAATGAT			
	* IL-10 *	F: CATGCTGCTGGGCCTGAA	[17]		
		R: CGTCTCCTTGATCTGCTTGATG			
Tight Junction Protein	*claudin 2*	F: CCTGCTCACCCTCATTGGAG	[7]	50.5 °C, 45 s	20 s
		R: GCTGAACTCACTCTTGGGCT			
	* ZO-1 *	F: CCGTAACCCCGAGTTGGAT	[7]		
		R: ATTGAGGCGGTCGTTGATG			
Zinc Importer	*ZIP-9*	F: CGTTCCATCTGCCTGCTGTC	[20]	49.4 °C, 45 s	20 s
		R: GCACCCAGAACAGTCACCAAC			
Antioxidant	*Zn/Cu-SOD-1*	F: GGCTTGTCTGATGGAGATCAT	XM_205064.1/[21]	60 °C, 30 s	3 s
		R: GCTTGCCTTCAGGATTAAAGTG			

**Table 3 animals-15-02373-t003:** Primer information for semi-quantitative real-time PCR of bacterial genes in the ceca in experiment 2.

	Primer Sequence (5′–3′)	Accession Number/Reference	Annealing	Extension
16S rRNA *Clostridium perfringens*	F: CGCATAACGTTGAAAGATGG	[22]	55 °C, 45 s	20 s
	R: CCTTGGTAGGCCGTTACCC			
*netB*	F: CGCTTCACATAAAGGTTGGAAGGC	[23]		
	R: TCCAGCACCAGCAGTTTTTCCT			
*cpa*	F: TGCATGAGCTTCAATTAGGT	[2]		
	R: TTAGTTTTGCAACCTGCTGT			

**Table 4 animals-15-02373-t004:** Growth performance of birds on day 21 in experiments 1 and 2.

			Day 21				
			BWG	FI	FCR	Lesion Score ^A^	Mortality ^B^
Experiment	Zinc Glycinate Concentration	Infection	g/bird	g/bird		(0–3)	%
1	40 mg/kg	Control	402.58 ± 28.95	639.71 ± 39.71	1.60 ^b^ ± 0.05	0.00 ^a^ (0–0)	0.0 ^a^ (0.30–7.71)
	40 mg/kg	NE	342.43 ± 16.03	656.76 ± 25.61	1.92 ^a^ ± 0.05	1.00 ^c^ (−1–−1.25)	8.3 ^b^ (4.20–20.47)
	80 mg/kg	NE	357.68 ± 13.94	683.59 ± 31.41	1.91 ^a^ ± 0.04	1.00 ^c^ (−1–−1.25)	0.0 ^a^ (0.30–7.71)
	120 mg/kg	NE	326.31 ± 17.93	602.68 ± 35.93	1.87 ^ab^ ±0.13	0.50 ^b^ (−0.33–−0.66)	4.2 ^ab^ (1.76–14.58)
	*p*-value		0.08	0.41	* <0.05	** <0.01	** <0.01
2	100 mg/kg	Control	546.34 ^a^ ± 25.55	929.54 ± 88.75	1.69 ± 0.09	0.00 ^a^ (0–0)	0.0 ^a^ (0.30–7.71)
	100 mg/kg	NE	416.03 ^b^ ± 21.25	759.18 ± 46.79	1.82 ± 0.05	1.11 ^c^ (−1–−1.25)	27.1 ^d^ (18.87–43.30)
	120 mg/kg	NE	481.76 ^ab^ ± 17.14	831.41 ± 24.03	1.74 ± 0.10	0.67 ^b^ (−0.42–−0.92)	14.6 ^bc^ (8.63–28.54)
	140 mg/kg	NE	427.53 ^b^ ± 34.46	751.92 ± 19.24	1.80 ± 0.10	0.67 ^b^ (−0.67–−0.67)	12.5 ^bc^ (7.08–25.92)
	160 mg/kg	NE	461.15 ^ab^ ± 27.60	779.44 ± 26.81	1.71 ± 0.06	0.72 ^b^ (−0.42–−0.92)	10.4 ^bd^ (5.60–23.24)
	*p*-value		* <0.05	0.09	0.74	** <0.01	** <0.01

Values in table are reported as mean ± SEM. ^A^ Lesion score is reported as median (interquartile range Q1–Q3). ^B^ Mortality is reported as percentage % ± confidence interval (CI). ^†^ Asterisk(s) denote the level of significance. ^a, b, c, d^ Means within the same column carrying different superscripts are significantly different at *p* < 0.05.

**Table 5 animals-15-02373-t005:** Growth performance of birds on day 28 in experiments 1 and 2.

			Day 28		
			BWG	FI	FCR
Experiment	Zinc Glycinate Concentration	Infection	g/bird	g/bird	
1	40 mg/kg	Control	608.71 ± 41.38	1062.31 ± 52.16	1.76 ^a^ ± 0.05
	40 mg/kg	NE	468.55 ± 19.13	1031.79 ± 36.70	2.22 ^b^ ± 0.11
	80 mg/kg	NE	509.46 ± 38.24	1104.24 ± 45.35	2.20 ^b^ ± 0.11
	120 mg/kg	NE	502.44 ± 49.41	1002.47 ± 36.14	2.06 ^b^ ± 0.15
	*p* value		0.09	0.4	* <0.05
2	100 mg/kg	Control	872.74 ^b^ ± 36.92	1204.39 ± 51.20	1.40 ^a^ ± 0.10
	100 mg/kg	NE	570.61 ^a^ ± 75.52	1229.10 ± 67.87	2.35 ^b^ ± 0.37
	120 mg/kg	NE	873.67 ^b^ ± 14.03	1389.87 ± 82.23	1.59 ^ab^ ± 0.10
	140 mg/kg	NE	808.53 ^ab^ ± 57.87	1217.64 ± 35.09	1.53 ^a^ ± 0.08
	160 mg/kg	NE	755.30 ^ab^ ± 93.07	1213.44 ± 76.31	1.67 ^ab^ ± 0.14
	*p* value		* <0.05	0.25	* <0.05

Values in table are reported as mean ± SEM. ^†^ Asterisk(s) denote the level of significance. ^a, b^ Means within the same column carrying different superscripts are significantly different at *p* < 0.05.

**Table 6 animals-15-02373-t006:** Pathohistological analysis of the jejunum on day 21 in experiment 1.

		Villi (µm)	Crypt (µm)	Villi to Crypt Ratio
Zinc Glycinate Concentration	Infection			
40 mg/kg	Control	362.91 ± 25.22	89.33 ^a^ ± 7.74	4.19 ^b^ ± 0.42
40 mg/kg	NE	309.58 ± 25.21	117.56 ^ab^ ± 4.95	2.65 ^a^ ± 0.23
80 mg/kg	NE	346.58 ± 11.64	127.55 ^bc^ ± 10.45	2.81 ^a^ ± 0.23
120 mg/kg	NE	332.83 ± 32.67	117.70 ^ab^ ± 7.94	2.85 ^a^ ± 0.42
*p*-value		0.51	** <0.01	** <0.01

Values are reported as mean ± SEM. ^a, b, c^ Means within the same column carrying different superscripts are significantly different at *p* < 0.05. ** means significance.

**Table 7 animals-15-02373-t007:** Two-way ANOVA of *Clostridium perfringens* growth curve as read by OD at 590 nm during co-culture with zinc at 0, 10, 100, or 1000 μM from zinc sulfate or zinc glycinate for 6 h.

		OD Values at Different Culture Time (Hours)						
Source	Concentration of Zinc μM	0	1	2	3	4	5	6
Control	0	0.05 ^bcd^	0.06 ^b^	0.13 ^ab^	0.29 ^ab^	0.52 ^ab^	0.78 ^ab^	0.75 ^b^
Sulfate	10	0.06 ^b^	0.06 ^b^	0.13 ^ab^	0.27 ^ab^	0.52 ^ab^	0.69 ^ab^	0.78 ^b^
Sulfate	100	0.06 ^bc^	0.06 ^b^	0.11 ^bc^	0.22 ^b^	0.44 ^b^	0.64 ^b^	0.74 ^b^
Sulfate	1000	0.05 ^d^	0.05 ^c^	0.07 ^d^	0.12 ^c^	0.26 ^c^	0.42 ^c^	0.57 ^c^
Glycinate	10	0.05 ^cd^	0.06 ^b^	0.12 ^bc^	0.29 ^a^	0.56 ^a^	0.76 ^ab^	0.81 ^ab^
Glycinate	100	0.05 ^d^	0.06 ^b^	0.09 ^cd^	0.23 ^b^	0.48 ^ab^	0.74 ^ab^	0.89 ^a^
Glycinate	1000	0.06 ^a^	0.07 ^a^	0.07 ^d^	0.08 ^c^	0.11 ^d^	0.26 ^d^	0.57 ^c^
Pooled SD		0.01	0.01	0.02	0.08	0.17	0.20	0.12
Probability								
ANOVA								
Source (S)		NS	*p* < 0.001	*p* < 0.01	NS	NS	NS	*p* < 0.01
Concentration (C)		*p* < 0.05	NS	*p* < 0.001	*p* < 0.001	*p* < 0.001	*p* < 0.001	*p* < 0.001
S × C		*p* < 0.001	*p* < 0.001	NS	*p* < 0.05	*p* < 0.001	*p* < 0.001	*p* < 0.001

^a, b, c, d^ Means carrying different superscripts are significantly different at *p* < 0.05.

## Data Availability

The original contributions presented in this study are included in the article. Further inquiries can be directed to the corresponding author(s).

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
