# Peer review of "Zinc Glycinate Alleviates Necrotic Enteritis Infection in Broiler Chickens"

_animals, 2025, doi:10.3390/ani15162373_

Round 1
Reviewer 1 Report
Comments and Suggestions for Authors
The article presents valuable information with consistent results, which is so important to comprehension of Zinc glycinate on necrotic enteritis infection in broiler chickens. My suggestions is solely to enhance the quality of the text.
Abstract
It can be improved and should include the study's conclusion and practical recommendations.
Materials and Methods
(Table S1) (Table S2)
Merging Tables S1 and S2 into a single, well-organized table and placing it directly in the Methodology section of the article instead of including it as supplementary material. Additionally, I suggest reviewing recently published articles in the journal to observe how similar tables have been presented
Tables 1 and 2
Please enhance the visual appearance of Tables 1 and 2 by formatting them neatly.
Results
Table 3
Dividing Table 3 into two separate tables to enhance visualization and improve readers' understanding of the results. One approach could be to separate the data based on the evaluated days (day 21 and day 28) or according to the experiments conducted (experiment 1 and experiment 2).
Line 227: Delete Table 3.
Line 242-244: On day 21 in experiment 2, infected birds supplemented with zinc glycinate at 120 , 140 and 160 mg/kg resulted in a lower lesion score than 100 mg/kg (P < 0.01)
(Table 3).
Line 244-247: I recommend revising the wording to enhance clarity.
Table 4:
Standardizing the format of Tables 3 and 4, particularly in the columns that identify the experimental groups.
In Table 3, using separate columns for "Zinc Glycinate Concentration" and "Infection" enhances clarity and allows for easier comparison between treatments.
However, in Table 4, the information regarding zinc concentration and infection status is presented together within the row text. This layout can impede direct reading and understanding of the groups.
Table 5:
Enhancing the formatting of the table for better clarity.
Discussion:
The final paragraph of the Discussion could be relocated to the Conclusion section, making minor adjustments if needed.
Additionally, the Discussion can be enhanced by more thoroughly exploring the immunological mechanisms related to zinc supplementation, as this was one of the primary objectives of the study.
Conclusion:
This section is currently too lengthy and detailed, resembling more of a Discussion. I recommend restructuring it to be more objective and concise, focusing only on the main findings and the most direct practical implications, without reintroducing the general disease context or justifying the study.
Author Response
Dear reviewers,
We would like to express our sincere gratitude to you and the reviewers for the thoughtful and constructive feedback on our manuscript. We appreciate the time and effort invested in reviewing our work and for the insightful suggestions that have helped us to improve the clarity, rigor, and overall quality of the manuscript.
We strive to accurately report our data while not making over-reaching conclusions. We have carefully considered all the comments and have revised the manuscript accordingly. Please see the detailed point-by-point reply to each comment, indicating the changes made and where they can be found in the revised manuscript.
We hope that the revisions address all your concerns raised and that the manuscript is now suitable for publication. Once again, thank you for your valuable feedback and for the opportunity to improve our work.
Sincerely,
Theros T. Ng
Comment 1: It can be improved and should include the study's conclusion and practical recommendations.
Response 1: Thank you for pointing this out. We have added the study’s conclusion and our practical recommendation in the abstract. This can be found on Page 1 Line 29 to 30 in the updated text in the manuscript.
Materials and Methods
Comment 2: (Table S1) (Table S2)
Merging Tables S1 and S2 into a single, well-organized table and placing it directly in the Methodology section of the article instead of including it as supplementary material. Additionally, I suggest reviewing recently published articles in the journal to observe how similar tables have been presented
Response 2: We have merged Tables S1 and S2 into Table 1. The new table can be found on line 91 in the updated manuscript.
We formatted the table according to the following publications:
Weaver AC, King WD, Verax M, Fox U, Kudupoje MB, Mathis G, Lumpkins B, Yiannikouris A. Impact of chronic levels of naturally multi-contaminated feed with Fusarium mycotoxins on broiler chickens and evaluation of the mitigation properties of different titers of yeast cell wall extract. Toxins. 2020 Oct 1;12(10):636.
Srinivasan R, Lumpkins B, Kim E, Fuller L, Jordan J. Effect of fiber removal from ground corn, distillers dried grains with solubles, and soybean meal using the Elusieve process on broiler performance and processing yield. Journal of Applied Poultry Research. 2013 Jul 1;22(2):177-89.
Comment 3: Tables 1 and 2
Please enhance the visual appearance of Tables 1 and 2 by formatting them neatly.
Response 3: We have adjusted the spacing of the tables to improve the visual appearance.
Results
Comment 4: Table 3
Dividing Table 3 into two separate tables to enhance visualization and improve readers' understanding of the results. One approach could be to separate the data based on the evaluated days (day 21 and day 28) or according to the experiments conducted (experiment 1 and experiment 2).
Response 4: We separated Table 3 into two separate tables as suggested. The new table (Table 4) can be found on line 251 in the updated manuscript.
Comment 5: Line 227: Delete Table 3.
Response 5: Removed.
Comment 6: Line 242-244: On day 21 in experiment 2, infected birds supplemented with zinc glycinate at 120 , 140 and 160 mg/kg resulted in a lower lesion score than 100 mg/kg (P < 0.01)
(Table 3).
Response 6: We have reworded this sentence. The revised line can be found on line 254 to 257 in the updated manuscript.
Comment 7: Line 244-247: I recommend revising the wording to enhance clarity.
Response 7: We have reworded this sentence. The revised line can be found on line 257 to 259 in the updated manuscript.
Comment 8: Table 4:
Standardizing the format of Tables 3 and 4, particularly in the columns that identify the experimental groups.
In Table 3, using separate columns for "Zinc Glycinate Concentration" and "Infection" enhances clarity and allows for easier comparison between treatments.
However, in Table 4, the information regarding zinc concentration and infection status is presented together within the row text. This layout can impede direct reading and understanding of the groups.
Response 8: Tables have been reformatted and standardized.
Comment 9: Table 5:
Enhancing the formatting of the table for better clarity.
Response 9: We have adjusted the spacing of this table and center aligned the text to improve clarity.
Discussion:
Comment 10: The final paragraph of the Discussion could be relocated to the Conclusion section, making minor adjustments if needed.
Response 10: We rewrote the conclusion based on other comments on the conclusion. Please let us know if the new version suffices.
Comment 11: Additionally, the Discussion can be enhanced by more thoroughly exploring the immunological mechanisms related to zinc supplementation, as this was one of the primary objectives of the study.
Response 11: We added the following: “We noticed improved viability of the intestinal cells in the presence of zinc, even as low as 10 µM. Cytotoxicity of the intestinal cells was also lowered by zinc. Our in vitro assays seem to indicate that zinc glycinate enhanced cellular proliferation and robustness overall, though zinc may not be enhancing particular cytokine responses.” Line 404-407
Conclusion:
Comment 12: This section is currently too lengthy and detailed, resembling more of a Discussion. I recommend restructuring it to be more objective and concise, focusing only on the main findings and the most direct practical implications, without reintroducing the general disease context or justifying the study.
Response 12: Please see response 10.
Reviewer 2 Report
Comments and Suggestions for Authors
Dear authors,
Your manuscript entitled “Zinc glycinate alleviates necrotic enteritis infection in broiler chickens” presents a valuable contribution to the field, addressing the important topic of dietary zinc supplementation in broiler chickens with the goal of mitigating the effects of necrotic enteritis. While the subject is relevant and the results are promising, several issues related to the content, structure, and data presentation must be addressed to improve the overall clarity and scientific rigor of the manuscript.
Below are detailed suggestions for revision:
Abstract
Please include essential methodological details such as the total number of chickens used, the number of chickens per treatment group, the number of replicates, the genetic line or hybrid of broilers used, and the duration of the feeding period with zinc glycinate-supplemented diets. The abstract is informative, but currently too difficult to read, insufficiently structured, and potentially overemphasized in one part.
We suggest rewriting the abstract by dividing it into logically structured sections: state the objective, briefly describe the experimental design and methods, summarize the most important results, and provide a concise conclusion.
Notably, the stated research objective in the abstract does not fully match the objective described in the Introduction. This discrepancy should be resolved.
Please note the following distinction:
There is a difference between assessing the effects of various zinc glycinate doses and determining the optimal dose. Dose optimization typically requires regression-based statistical modeling, which is not used in this study.
In your manuscript, you assessed the most effective dose relative to the NRC recommendation of 100 mg/kg by increasing and decreasing zinc concentrations, then identifying the most effective level through statistical comparisons. This is valid, but please be precise in defining the scope and limitations of your experimental design.
Materials and Methods
Clearly state the number of birds per group, then indicate the number of replicates and the number of birds per replicate.
The method section lacks a description of how the feed conversion ratio (FCR) was calculated. Additionally, clarify when and how body weight and feed intake were measured, and whether feed refusals were accounted for.
Please explain why the effects of zinc glycinate were only monitored between days 21–28 of the trial.
Since broiler production typically lasts up to 42 days, we assume you also reared the birds until that age. If available, performance data until day 42 would be highly informative and should be included.
Results and Discussion
There is no statistical or analytical integration of gene expression data with phenotypic outcomes (e.g., lesion score, body weight gain, histology). This limits the biological interpretation of gene expression changes.
Consider adding correlation analyses or at least a discussion linking molecular and clinical results more directly.
Conclusion
The conclusion ends with a generalized statement:
“These results suggest zinc concentration, rather than source, is key in modulating host cell responses during infection.”
This is a strong claim, yet the manuscript does not provide a direct comparison of different zinc sources—zinc glycinate was the primary form used, with zinc sulfate tested only in limited in vitro assays. We recommend either softening this statement or integrating supporting rationale more clearly into the main text.
Style and Language Corrections
Lines 25–29: The sentence structure is awkward. The term "cytotoxicity" is repeated multiple times. Please revise for conciseness and readability.
Line 88: Spelling error – “previoulsy” should be corrected to “previously.”
Author Response
Dear reviewers,
We would like to express our sincere gratitude to you and the reviewers for the thoughtful and constructive feedback on our manuscript. We appreciate the time and effort invested in reviewing our work and for the insightful suggestions that have helped us to improve the clarity, rigor, and overall quality of the manuscript.
We strive to accurately report our data while not making over-reaching conclusions. We have carefully considered all the comments and have revised the manuscript accordingly. Please see the detailed point-by-point reply to each comment, indicating the changes made and where they can be found in the revised manuscript.
We hope that the revisions address all your concerns raised and that the manuscript is now suitable for publication. Once again, thank you for your valuable feedback and for the opportunity to improve our work.
Sincerely,
Theros T. Ng
Comment 1: Please include essential methodological details such as the total number of chickens used, the number of chickens per treatment group, the number of replicates, the genetic line or hybrid of broilers used, and the duration of the feeding period with zinc glycinate-supplemented diets. The abstract is informative, but currently too difficult to read, insufficiently structured, and potentially overemphasized in one part.
Response 1: We added the number of birds, genetic line, duration, and biological replicates in the abstract. Line 20-21
Comment 2: We suggest rewriting the abstract by dividing it into logically structured sections: state the objective, briefly describe the experimental design and methods, summarize the most important results, and provide a concise conclusion.
Response 2: We added the previously mentioned details and included all the information requested with P values while limiting the word count to 200 per the journal guidelines. We added the last sentence as the conclusion of the study in the abstract. Line 29-30
Comment 3: Notably, the stated research objective in the abstract does not fully match the objective described in the Introduction. This discrepancy should be resolved.
Please note the following distinction:
There is a difference between assessing the effects of various zinc glycinate doses and determining the optimal dose. Dose optimization typically requires regression-based statistical modeling, which is not used in this study.
In your manuscript, you assessed the most effective dose relative to the NRC recommendation of 100 mg/kg by increasing and decreasing zinc concentrations, then identifying the most effective level through statistical comparisons. This is valid, but please be precise in defining the scope and limitations of your experimental design.
Response 3: We avoided the language of determining optimal dose/concentration throughout our manuscript. We did not use regression-based statistical analysis due to the sample size and study design.
- It is not a complete factorial design. Mainly because we are only focused on zinc during the infection. Zinc concentration in healthy animals is beyond the scope of our study. Infectious studies like ours and a complete factorial design experiment would be too cost-prohibitive.
- We have 6 biological replicates, which add up to only 24 or 30 observations in each experiment. The statistical power for regression analysis is low. Even if we get a decent R2, it would be misleading.
- Experiments 1 and 2 are not a direct repeat of each other.
In summary, we think that the word effective concentration balances both worlds without exaggerating what our findings can conclude. Please let us know if further changes are needed.
Materials and Methods
Comment 4: Clearly state the number of birds per group, then indicate the number of replicates and the number of birds per replicate.
Response 4: We included the following in lines 82-84 in the revised manuscript.
Comment 5: The method section lacks a description of how the feed conversion ratio (FCR) was calculated. Additionally, clarify when and how body weight and feed intake were measured, and whether feed refusals were accounted for.
Response 5: We added the following in lines 111 to 115 in the revised manuscript.
“Birds and feed were weighed on days 0, 14, 21, and 28. Body weight and feed intake were measured on a per-cage (replicate) basis. Feed intake was calculated by subtracting the weight of feed refusals from the total feed provided. Feed conversion ratio (FCR) was calculated for each replicate as the total feed intake divided by the total body weight gain, and averaged per bird within each cage.”
Comment 6: Please explain why the effects of zinc glycinate were only monitored between days 21–28 of the trial.
Since broiler production typically lasts up to 42 days, we assume you also reared the birds until that age. If available, performance data until day 42 would be highly informative and should be included.
Response 6: Unfortunately, we did not conduct this study beyond 28 days. As you correctly pointed out that broiler production can last up to 42 days. They can be slaughtered as early as 28 days. The private research organization (Southern Poultry Research Inc.), which conducted the animal trials, recommended this timeline based on their experimental infection model and other studies that were ongoing.
Bortoluzzi C, Lumpkins B, Mathis GF, França M, King WD, Graugnard DE, Dawson KA, Applegate TJ. Zinc source modulates intestinal inflammation and intestinal integrity of broiler chickens challenged with coccidia and Clostridium perfringens. Poultry science. 2019 May 1;98(5):2211-9.
Bortoluzzi C, Vieira BS, Lumpkins B, Mathis GF, King WD, Graugnard D, Dawson KA, Applegate TJ. Can dietary zinc diminish the impact of necrotic enteritis on growth performance of broiler chickens by modulating the intestinal immune-system and microbiota?. Poultry science. 2019 Aug 1;98(8):3181-93.
Regardless, we added this to explain the limitations in the discussion. Please see line 418 to 425.
“This study focused on the early growth phase and disease impact, concluding at day 21 to capture the acute effects of necrotic enteritis and early performance outcomes. While broilers are commonly raised to day 42, early slaughter is not uncommon in commercial operations, particularly under disease pressure. Future studies are warranted to assess the long-term effects of zinc supplementation on performance and health through to market age.”
Results and Discussion
Comment 7: There is no statistical or analytical integration of gene expression data with phenotypic outcomes (e.g., lesion score, body weight gain, histology). This limits the biological interpretation of gene expression changes.
Consider adding correlation analyses or at least a discussion linking molecular and clinical results more directly.
Response 7: We chose not to statistically integrate gene expression data with phenotypic outcomes such as lesion scores or body weight gain due to the limited scope of our gene expression panel and the inherent complexity of immune signaling. The cytokines we assessed have pleiotropic and context-dependent functions, making it difficult to draw direct or causal relationships between their expression at a single time point and multifactorial outcomes. Moreover, gene expression reflects only one layer of biological regulation and does not always correlate linearly with phenotype, particularly in the context of acute immune responses. Therefore, we felt it would be an over interpretation to attempt integration without more comprehensive temporal and mechanistic data. Future studies employing broader transcriptomic profiling may be better suited to explore such relationships.
Conclusion
Comment 8: The conclusion ends with a generalized statement:
“These results suggest zinc concentration, rather than source, is key in modulating host cell responses during infection.”
This is a strong claim, yet the manuscript does not provide a direct comparison of different zinc sources—zinc glycinate was the primary form used, with zinc sulfate tested only in limited in vitro assays. We recommend either softening this statement or integrating supporting rationale more clearly into the main text.
Response 8: The conclusion has been rewritten based on reviewer 1’s comment. This sentence is no longer present.
Comment 9: Style and Language Corrections
Lines 25–29: The sentence structure is awkward. The term "cytotoxicity" is repeated multiple times. Please revise for conciseness and readability.
Response 9: Thank you. We reworded this sentence.
Comment 10: Line 88: Spelling error – “previoulsy” should be corrected to “previously.”
Response 10: The spelling has been corrected.
Reviewer 3 Report
Comments and Suggestions for Authors
Zinc Glycinate Alleviates Necrotic Enteritis Infection in Broiler Chickens
Dear Authors,
the manuscript is interesting and quite well-prepared. The conclusions regarding increasing zinc concentration to 120 mg/kg are perfectly reasonable in reducing losses associated with necrotic enteritis in chickens. However, the main element that needs improvement, in my opinion, is the description of the significance of differences between treatments in the Tables and Figures, and the inclusion of a table with a two-way ANOVA test and a description of the significance of differences separately for each experimental factor, or the addition of p values for one-way ANOVA. Besides this revision, a point-of-view edition of the text will also be required.
Below I added some suggestions helpful in revision process:
Lines 22-29
Data was gathered as a sample from population in this case p-value must be used instead of P-value.
Lines 51-62
Line spacing in this paragraph must be adapted to the rest of text or the rest text to this part.
Line 55
Eimeria sp. or spp. ?
Lines 167, 178, 185, 195 and 205
CO2 (form with subscript can be used).
Line 215
Information about Shapiro-Wilk’s test can be added in this case to confirmation normal distribution of data in each treatment.
Line 227
Table 3 can be mentioned on the end of first sentence ‘… concentrations (Table 3).’ instead of subsection title.
Lines 236-238
The same as in lines 22-29.
Please check to the line 329.
Line 238
Table 3
Title of Table is not part of it (please delete top border).
Please check significance level in experiment 1, Lesion score, in this form 0.50 should have b letter, and 1.00 c letter.
But normally a letter is added to the highest value, and then following letter/s are used.
p-value instead of P-value must be used.
p-value can be specified with three decimals.
Please check significance level also in case of Mortality in second experiment, because taking into consideration linear course of numerical values 10.4 should have ab letters, 15.5bc, 14.6c , and 27.1d.
Description of significance level can be also added below the Table 3 (letters in superscript and asterisk/s).
Maybe capital letters can be used in case of description of estimators do differentiate it from significance level.
Line 251
Table 4
Title of Table is not part of it (please delete top border).
Description below the table can be shortened to values in table are reported as mean ± SEM (without a letter).
Please check description of significant differences between treatments using letters in superscripts to Crypt column (linear order).
Line 270
Figure 1
Description of significance level can be also added on the end of line 270 (letters a,b significant at p<0.05 ).
Title/Description of Figure 1 must be adapted to left margin line.
Line 276
Figure 2
Description of significance level can be also added on the end of line 276 (letters a,b significant at p<0.05 ).
Title/Description of Figure 2 must be adapted to left margin line.
Line 282
Figure 3
Please check if there are significant differences between treatments in case of each variable.
If not description on the end of line 285 can be added: ‘No significant differences between treatments (p>0.05).
Title/Description of Figure 3 must be adapted to left margin line.
Line 297
Figure 4a, 4b
Description of significance level can be also added on the end of line 299 (letters a,b significant at p<0.05 ).
Title/Description of Figure 4 must be adapted to left margin line.
Line 309
Figure 5a, 5b
Description of significance level can be also added on the end of line 299 (letters a, b, c, d significant at p<0.05 ).
Title/Description of Figure 5 must be adapted to left margin line.
Please check description of significant differences between treatments using letters in Figure 5a (linear order, lowest value in case of Sulfate 100 µM).
Line 325
Figure 6
Title/Description of Figure 6 must be adapted to left margin line.
Description of significance level can be also added on the end of line 327 (* describes significant differences between treatments at p<0.05? ).
Line 328
Table 5
Title of Table is not part of it (please delete top border).
Description of significance level is conducted to one-way ANOVA, maybe it will be worth to add p-value below Pooled SD row (SEM value calculated for all replications can be also used).
Three decimals can be present, especially in 0 hour, because also description of significant differences between treatments is confused. Generally, in 0 hour there should not be significant differences, but probably concentration of source can have influence for measure value after inoculation by bacteria.
In case of p-value 3 decimals maximally can be presented.
Lines 400-416
Clostridium perfringens and C. perfringens (italics must be used).
Line 444
Table S1
Title of Table is not part of it (please delete top border).
Soybean meal (48 % CP), %.
Vegetable fat can be more precise determined: Soybean oil (more probable) or Rapeseed oil.
Line 445
Table S2
Title of Table is not part of it (please delete top border).
Lines 470-580
References section
References must be adapted to pattern described in Instructions for Authors: https://www.mdpi.com/journal/animals/instructions
Abbreviation/s of Journal’s name must be added.
Doi link/number must be added on the end of each reference for each was generated.
I.e.:
- Al-Sheikhly, F.; Al-Saieg, A. Role of coccidia in the occurrence of necrotic enteritis of chickens. Avian Dis. 1980, 24(2), 324-333. https://doi.org/10.2307/1589700
Author Response
Dear reviewers,
We would like to express our sincere gratitude to you and the reviewers for the thoughtful and constructive feedback on our manuscript. We appreciate the time and effort invested in reviewing our work and for the insightful suggestions that have helped us to improve the clarity, rigor, and overall quality of the manuscript.
We strive to accurately report our data while not making over-reaching conclusions. We have carefully considered all the comments and have revised the manuscript accordingly. Please see the detailed point-by-point reply to each comment, indicating the changes made and where they can be found in the revised manuscript.
We hope that the revisions address all your concerns raised and that the manuscript is now suitable for publication. Once again, thank you for your valuable feedback and for the opportunity to improve our work.
Sincerely,
Theros T. Ng
Comment 1:
Lines 22-29
Data was gathered as a sample from population in this case p-value must be used instead of P-value.
Response: Changed
Comment 2: Lines 51-62
Line spacing in this paragraph must be adapted to the rest of text or the rest text to this part.
Response 2: Corrected
Comment 3: Line 55
Eimeria sp. or spp. ?
Response 3: This no longer exists in the revised manuscript.
Comment 4: Lines 167, 178, 185, 195 and 205
CO2 (form with subscript can be used).
Response 4: The subscripts have been corrected.
Comment 5: Line 215
Information about Shapiro-Wilk’s test can be added in this case to confirmation normal distribution of data in each treatment.
Response 5: We did not perform this test.
Comment 6: Line 227
Table 3 can be mentioned on the end of first sentence ‘… concentrations (Table 3).’ instead of subsection title.
Response 6: We made table 3 into separate tables (4 and 5) as recommended by reviewer 1. Tables 4 and 5 are referenced on line 238.
Comment 7: Lines 236-238
The same as in lines 22-29.
Please check to the line 329.
Line 238
Table 3
Title of Table is not part of it (please delete top border).
Response 7: We corrected the top border of the tables.
Comment 8: Please check significance level in experiment 1, Lesion score, in this form 0.50 should have b letter, and 1.00 c letter.
But normally a letter is added to the highest value, and then following letter/s are used.
Response 8: We have corrected letters for significant values by the order of the values.
Comment 9: p-value instead of P-value must be used.
Response 9: Changed
Comment 10: p-value can be specified with three decimals.
Response 10: We have limited all p-values to 3 decimals or less.
Comment 11: Please check significance level also in case of Mortality in second experiment, because taking into consideration linear course of numerical values 10.4 should have ab letters, 15.5bc, 14.6c , and 27.1d.
Response 11: We have corrected letters for significant values by the order of the values.
Comment 12: Description of significance level can be also added below the Table 3 (letters in superscript and asterisk/s).
Response 12: We added a description of the significance level in all figures.
Comment 13: Maybe capital letters can be used in case of description of estimators do differentiate it from significance level.
Response 13: We used capitalized letters for captions. And non-capitalized letters for significance level.
Comment 14: Line 251
Table 4
Title of Table is not part of it (please delete top border).
Response 14: We corrected the top border of the tables.
Comment 15: Description below the table can be shortened to values in table are reported as mean ± SEM (without a letter).
Response 15: In the description below the table has been shortened.
Comment 16: Please check description of significant differences between treatments using letters in superscripts to Crypt column (linear order).
Response 16: We have corrected letters for significant values by the order of the values.
Comment 17: Line 270
Figure 1
Description of significance level can be also added on the end of line 270 (letters a,b significant at p<0.05 ).
Response 17: Description of significance level has been added.
Comment 18: Title/Description of Figure 1 must be adapted to left margin line.
Response 18: Figure description has been adapted to left margin line.
Comment 19: Line 276
Figure 2
Description of significance level can be also added on the end of line 276 (letters a,b significant at p<0.05 ).
Response 19: Description of significance level has been added.
Comment 20: Title/Description of Figure 2 must be adapted to left margin line.
Response 20: Figure description has been adapted to left margin line.
Comment 21: Line 282
Figure 3
Please check if there are significant differences between treatments in case of each variable.
If not description on the end of line 285 can be added: ‘No significant differences between treatments (p>0.05).
Response 21: No significant differences between treatments (p>0.05) has not been added
Comment 22: Title/Description of Figure 3 must be adapted to left margin line.
Response 22: Figure description has been adapted to left margin line.
Comment 23: Line 297
Figure 4a, 4b
Description of significance level can be also added on the end of line 299 (letters a,b significant at p<0.05 ).
Response 23: Description of significance level has been added.
Comment 24: Title/Description of Figure 4 must be adapted to left margin line.
Response 24: Figure description has been adapted to left margin line.
Comment 25: Line 309
Figure 5a, 5b
Description of significance level can be also added on the end of line 299 (letters a, b, c, d significant at p<0.05 ).
Response 25: Description of significance level has been added.
Comment 26: Title/Description of Figure 5 must be adapted to left margin line.
Response 26: Figure description has been adapted to left margin line.
Comment 27: Please check description of significant differences between treatments using letters in Figure 5a (linear order, lowest value in case of Sulfate 100 µM).
Response 27: We have corrected letters for significant values by the order of the values.
Comment 28: Line 325
Figure 6
Title/Description of Figure 6 must be adapted to left margin line.
Response 28: Figure description has been adapted to left margin line.
Comment 29: Description of significance level can be also added on the end of line 327 (* describes significant differences between treatments at p<0.05? ).
Response 29: Description of significance level has been added. Significant at different levels but at least 0.01.
Comment 30: Line 328
Table 5
Title of Table is not part of it (please delete top border).
Response 30: Table margin has been corrected.
Comment 31: Description of significance level is conducted to one-way ANOVA, maybe it will be worth to add p-value below Pooled SD row (SEM value calculated for all replications can be also used).
Response 31: Table 7 has been updated. Table 7 is two-way ANOVA. The in vivo experiments were analyzed using one-way ANOVA because of the incomplete factorial design. The in vitro experiments were analyzed using two-way ANOVA. p-values were included in the table.
Comment 32: Three decimals can be present, especially in 0 hour, because also description of significant differences between treatments is confused. Generally, in 0 hour there should not be significant differences, but probably concentration of source can have influence for measure value after inoculation by bacteria.
Response 32: They are different at 0 hour, however. That is the reason we reported it as is. The difference is small, but zinc could have changed the color without the bacteria themselves.
Comment 33: In case of p-value 3 decimals maximally can be presented.
Response 33: We have limited all p-values to 3 decimals.
Comment 34: Lines 400-416
Clostridium perfringens and C. perfringens (italics must be used).
Response 34: Corrected
Comment 35: Line 444
Table S1
Title of Table is not part of it (please delete top border).
Response 35: Top border has been corrected.
Comment 36: Soybean meal (48 % CP), %.
Vegetable fat can be more precise determined: Soybean oil (more probable) or Rapeseed oil.
Response 36: We checked Southern Poultry Research Inc.’s document. It is Soybean oil and it has been added.
Comment 37: Line 445
Table S2
Title of Table is not part of it (please delete top border).
Response 37: Top border has been corrected.
Comment 38: Lines 470-580
References section
References must be adapted to pattern described in Instructions for Authors: https://www.mdpi.com/journal/animals/instructions
Response 38: MDPI reference format is used in Endnote.
Comment 39: Abbreviation/s of Journal’s name must be added.
Response 39: Abbreviations of journals have been added
Comment 40: Doi link/number must be added on the end of each reference for each was generated.
I.e.:
Al-Sheikhly, F.; Al-Saieg, A. Role of coccidia in the occurrence of necrotic enteritis of chickens. Avian Dis. 1980, 24(2), 324-333. https://doi.org/10.2307/1589700
Response 40: DOIs have been added except for an old study by Long and Truscott that I could not find the DOI and some online materials (Aviagen), books, and thesis (Luoma) do not have DOI.
Round 2
Reviewer 2 Report
Comments and Suggestions for Authors
Dear authors,
you have successfully corrected the paper according to the instructions. The corrections have certainly improved the paper.
I recommend the paper for further procedure.
Reviewer
Author Response
Dear Reviewer,
Thank you for your feedback.
Yours sincerely,
Theros Ng
Reviewer 3 Report
Comments and Suggestions for Authors
Dear Authors,
Thank you for the Revision process, this time I have only a few suggestions mainly from the point of view edition of text.
Table 4
Maybe better will be to change orientation of this page to fit entire table to width of page (same as in case of Table 2).
In experiment 2 in case of value in row with 120 mg/kg NE, value 14.6 is emphasized with bc (b must be only changed).
Line 475
In case of references 2, 15, 18, 29 abbreviation/s of Journal’s name must be applied.
Line 587
Second part of doi link/number must be moved to the line 586.
Author Response
Comment 1: Table 4
Maybe better will be to change orientation of this page to fit entire table to width of page (same as in case of Table 2).
Response 1: The orientation has been changed.
Comment 2: In experiment 2 in case of value in row with 120 mg/kg NE, value 14.6 is emphasized with bc (b must be only changed).
Response 2: The superscript has been changed.
Line 475
Comment 3: In case of references 2, 15, 18, 29 abbreviation/s of Journal’s name must be applied.
Response 3: After checking comment 3, weren’t able to find the error. We found that references 15, 18, and 29 (please see screenshot below), the journal names were abbreviated. Perhaps the reviewer was viewing the pre-markup version. We were not able to find the specific error. We apologize for the error. However, we have rechecked all the references, including the journal abbreviation and DOIs.
Comment 4: Line 587
Second part of doi link/number must be moved to the line 586
Response 5: The last line was line 580. We were not able to find the specific error. We apologize for the error. However, we have rechecked all the references, including the journal abbreviation and DOIs.
